

# Assessment of uncertainties on stage-discharge rating curves: A large scale application to Québec hydrometric network

Alain Mailhot[1], Guillaume Talbot[1], Samuel Bolduc[1], Claudine Fortier[2]

[1]Institut national de la recherche scientifique, Centre Eau, Terre et Environnement, Québec City, G1K 9A9, Québec (Canada)

[2]Direction de l'hydrométrie et des prévisions hydrologiques, Ministère de l'Environnement, de la Lutte contre les changements climatiques, de la Faune et des Parcs, Québec City, G1R 5V7, Québec (Canada)

*Correspondence to*: Alain Mailhot (alain.mailhot@inrs.ca)

**Abstract.** Rating curves (RC), which establish a relationship between stage and discharge at a given cross-section of a river, are largely used by national agencies to measure flow. RC are constructed from gauging measurements and are usually represented by power functions, a mathematical function frequently used to represent stage-discharge relationship of standard hydraulic structures. Uncertainties on estimated flows based on rating curves can be significant, especially for high and low flow regimes. It is therefore important to report these uncertainties as accurately as possible. Many approaches estimating the sources of uncertainties on flows have been proposed but are generally too complex for large scale application to hydrometric networks. This paper proposed an approach to develop rating curves and assess the corresponding uncertainties on estimated flow that can be readily applied to large-scale hydrometric networks. This approach takes into consideration possible changes in RC over time due to hydraulic or geomorphologic modifications and assessed if one or two power functions are needed to adequately represent the stage-discharge relationship over the available range of gauged stages. RC at Quebec hydrometric stations have been constructed. Relative differences between flows estimated from the RC and gauged flows are used to assess uncertainties on estimated flow. They were adjusted to normal or logistic distributions with constant (stage-independent uncertainties) or stage-dependent scale parameters (stage-dependent uncertainties). Mean standard deviation on estimated flows for RC with stage-independent uncertainties (75.0% of the RC) is 6.5%, while for RC with stage-dependent uncertainties, they increase significantly at low stages reaching values larger than 20% for some RC at the lowest gauged stage.

## 1 Introduction

Stage-discharge rating curves (RC), which establish a relationship between stage and discharge at a given cross-section of a river, are largely used by national agencies to assess flow since stage is much more easily measured than flow (WMO 2010a; Le Coz *et al*. 2014; Fenton 2018; McMahon and Peel, 2019). These data are crucial to assess the recurrence of peak flow, the development of flood maps and, more generally, to provide information on the watercourse hydraulic and hydrological regimes. However, uncertainties on estimated flows can be substantial, especially for high and low flow regimes.




Considering the practical importance of these data, it is therefore essential to accurately report these values and provide corresponding uncertainties (McMillan *et al*. 2017), which is generally not done (Herschy, 2002; Hamilton and Moore, 2012). Many approaches have been proposed in the literature to estimate the various sources of uncertainties on flows (for a review see Kiang *et al*. 2018; see also Table 1 in McMahon and Peel, 2019). However, most of the approaches are not

adapted or too complex for large scale application to hydrometric networks.

The use of RC is based on several hypotheses about the hydraulic conditions prevailing at the gauging site, and the existence of stable hydraulic controls. This includes, among others, the absence of backwater effects, constant channel roughness, smooth geometry of the cross-section, steady-state flows (Rantz 1982a, 1982b). Departures from these conditions may results in uncertainties and bias on estimated flows. For example, unsteady flow conditions may lead to hysteresis effects.

Thus, using RC for stations located in reaches with small channel slopes under rapid increasing flows may lead to significant bias on estimated discharges (Perret *et al*. 2022).

Various types of uncertainties must be accounted for when estimating discharges from RC (Le Coz *et al*. 2014; Di

Baldassarre and Montanari 2009; McMillan *et al*. 2018): 1) uncertainties on measured stages and discharges; 2) adequacy of RC to represent the stage-discharge relationship across all measured stages (interpolated part of the RC), and for water level outside the gauged range (extrapolated part of the RC); 3) possible changes in stage-discharge relationship over time due to geomorphological changes or sedimentation (Morlot *et al*. 2014; Mansanarez *et al*. 2019a); 4) seasonal changes due to vegetation growth (Perret *et al*. 2021); 5) stage-discharge hysteresis due to transient flow (Perret *et al*. 2022). Various

methods have been developed and proposed to take into consideration these uncertainties.

Relative contributions of these uncertainties depend on the stage, flow conditions, type of hydraulic control, and characteristics of the section or the reaches controlling the stage-discharge relationship. McMillan *et al*. (2012) have provided some benchmark values for the various contributions to discharge uncertainties. It is generally assumed that stage

uncertainties are relatively small (less than ± 10 mm according to McMillan *et al*. 2012) and therefore usually neglected. Uncertainties on flow measurements are larger and mainly depends on measurement techniques. McMillan *et al*. (2012) report uncertainties less than 20% on measured flows for the commonly used velocity-area method (Pelletier 1988) and less than around 5% for acoustic Doppler current profiling method.

Two main representations of RC have been proposed. The first one is a power function (e.g., Le Coz *et al*. 2014):

$$Q_{RC} = a\,(h - b)^c \tag{1}$$



where $Q_{RC}$ is the discharge (m³/sec), h is the water level relative to a datum (m), while a, b, and c are three calibration parameters. The exponent c is related to the type of hydraulic control and is hereafter called hydraulic exponent. Such simple generic equation can be derived, under specific assumptions, from formulas for uniform flows (Chézy, Manning-Strickler), and from usual hydraulic structures (weirs, gauging flumes, etc.; Le Coz *et al*. 2014; Le Coz *et al*. 2011). The power function is considered in this study.

The second one is based on interpolation functions (e.g., cubic splines as in Hrafnkelsson *et al*. 2012; or Chebyshev polynomials as in McMahon and Peel, 2019). Fenton (2018), promoting the use of such approach, argued that the power function is too simplistic and likely over-simplified the complexity of the hydraulics underlying the stage-discharge relationship at many cross-sections. Despite these legitimate criticisms, wide application of the power function has shown that stage-discharge curves are surprisingly well-represented by such functions, which is the case in the actual application.

Once the mathematical representation is selected, many issues must be considered in the development of RC and the evaluation of uncertainties of estimated flows. The first one relates to possible changes in stage-discharge relationship through time due, for instance, to geomorphologic changes in cross-section. Non-stationary stage-discharge relationship implies the use of different RC over different time periods, or even of RC with time-dependent parameters (Morlot *et al*. 2014; Mansanarez *et al*. 2019a). Using an inadequate RC over some periods may result in major bias in estimated flows.

The second major issue relates to changes in control sections when stages crosses specific thresholds associated for instance to major changes in the profile of the control sections (e.g., transition to flood plain) or in the nature of the hydraulic controls (WMO 2010a, 2010b). Such changes will manifest through modifications in the shape of the stage-discharge relationship, which means that more than one power function must be used to represent the stage-discharge relationship over the whole range of gauged stage. These modifications may result from changes in hydraulic control. It is also one of the main limitations of the stage-discharge relationship since changes in hydraulic control occurring for instance at low or high discharges may not be captured by available gauging measurements. This is an important issue in Quebec considering that annual maximum flows and flooding generally occur in spring (snowmelt or rain-on-snow events) and that gauging in these conditions may be difficult and hazardous.

The main objective of this paper is to develop RC and assess the corresponding uncertainties on estimated flow based on an approach that can be readily applied to hydrometric networks. Such approach should deal with possible changes in RC over time and assessed the number of power functions needed to adequately represent the RC over the whole range of gauged stages. The resulting approach is applied to the Quebec hydrometric network.





The paper is structured as follow. Section 2 describes the available datasets and presents some preliminary analysis. Section 3 presents the different steps of the approach to develop the RC, first explaining how RC are adjusted to available gaugings (Section 3.1), then the procedure to partition the initial gauging period into subperiods, each one represented by a RC

100 (Section 3.2), and finally the procedure to determine if one or two power functions are needed to represent each RC (Section 3.3). Results of the application of the proposed procedures to the Quebec hydrometric network are detailed in Section 4. Uncertainty models for estimated discharges from RC are presented in Section 5 with corresponding results for Quebec hydrometric network. Section 6 presents the conclusions and provides perspectives for future work.

## 2 Available datasets

105 A total of 173 hydrometric stations operated by the *Ministère de l'Environnement, de la Lutte contre les changements climatiques, de la Faune et des Parcs* (MELCCFP) in Québec (Canada) were considered. Available gaugings, stages with corresponding flows, were compiled at each station. Only gaugings in open-water conditions were considered (no ice cover, flows influenced by ice). Also, gaugings presumably affected by backflows or obstructions were eliminated. Since most rivers in Quebec are partially or totally covered by ice in winter, gaugings in open-water conditions are mainly done from

110 April to November (Figure 1) and gauging campaigns are often carried out in April and May, during spring peak flows associated to snowmelt, which usually correspond to the annual maximum flow recorded. A total of 10 087 gaugings were considered, for an average of 58.3 gaugings per station, covering periods ranging from 3 to 98 years. Stations with less than 6 gaugings were discarded.



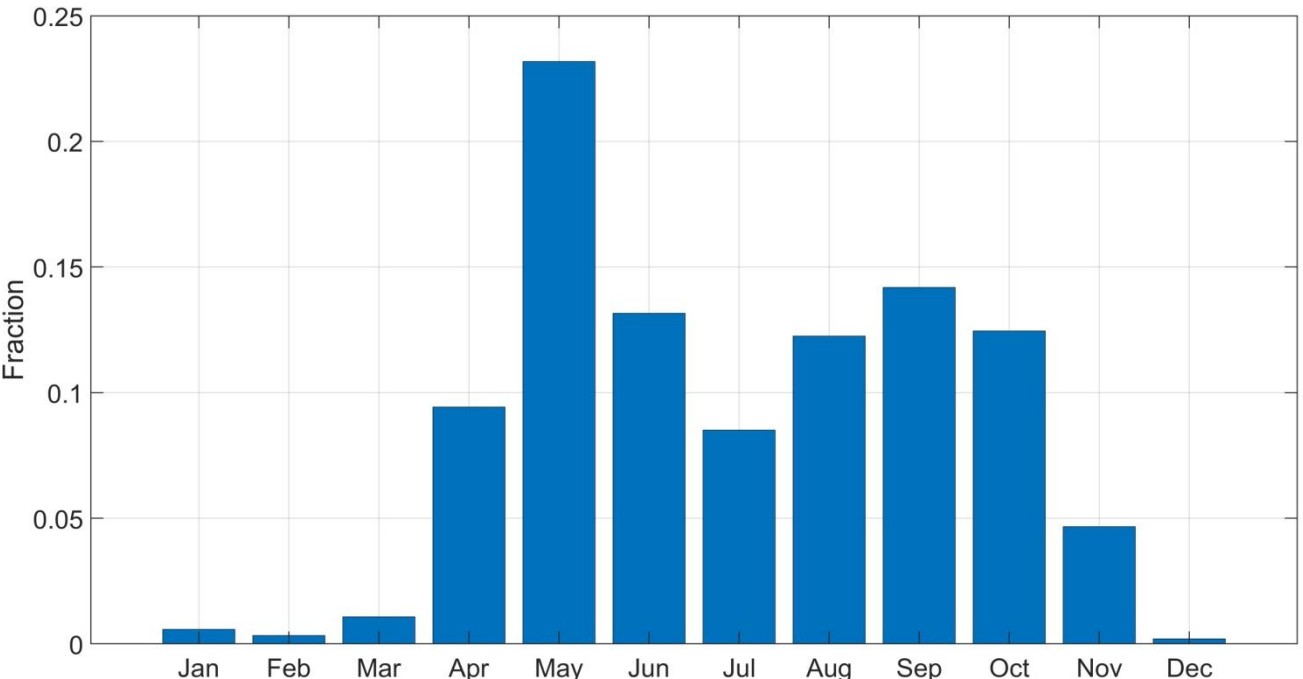

**Figure 1: Monthly distribution of open-water gaugings at the 173 hydrometric stations under study.**

## 3 General Approach

The proposed approach proceeds in three steps: 1) adjustment of a single power function to all gaugings available at a given station; 2) segmentation of the original gauging sequence (GS) into sub-sequences to account for possible changes in RC over time; 3) adjustment of power function to each GS. The following sections further details each of these steps.

### 3.1 Adjustment of the rating curve

Gauging measurements at a given station are represented by $(h_j, Q_j)$ with $h_j$ the measured water levels and $Q_j$ the corresponding discharges. Parameters of the power function (Eq. 1) used to represent the RC, are estimated by minimizing the Mean Square Relative Error (MSRE) defined by:

$$MSRE = \frac{1}{N}\sum_{j=1}^{N}\left[\frac{Q_{RC}(h_j;a,b,c)-Q_j}{Q_j}\right]^2 \qquad (2)$$





where N is the number of gauging measurements and $Q_{RC}(h_j; a, b, c)$ the estimated discharge from the RC. Relative errors on estimated discharge were considered since it seems reasonable to assume that they are independent of discharge (this assumption will be further investigated). Parameters of the RC were estimated using the Nelder-Mead non-linear optimisation algorithm (Lagarias et al. 1998). RC are adjusted if six or more gaugings are available.

More than one power function is needed to represent the stage-discharge relationship in some cases (section 3.3 further discusses this point). RC is therefore represented in these cases by two power functions according to:

$$Q(h) = \begin{cases} Q_1(h) = a_1 \ (h - b_1)^{c_1} & h \leq h' \\ Q_2(h) = a_2 \ (h - b_2)^{c_2} & h \geq h' \end{cases} \tag{3}$$

with $(a_1, b_1, c_1)$ and $(a_2, b_2, c_2)$ the parameters of the two power functions associated to the water levels below and above $h'$, hereafter called the transition level. Continuity at $h = h'$ is imposed by setting $Q_1(h') = Q_2(h')$. The MSRE (Eq. 2) is therefore minimized by finding the values of $(a_1, b_1, c_1)$, $(a_2, b_2, c_2)$, and $h'$ satisfying the constraint $Q_1(h') = Q_2(h')$.

### 3.2 Temporal partition of the gauging series

Changes in RC resulting from modifications of hydraulic or geomorphologic conditions over time were assessed following the procedure presented in Fig. 2. RC were first estimated using all gaugings at a station. Resulting RC are hereafter called baseline RC, as in Darienzo et al. (2021). The analysis is performed when six gaugings or more are available, which is the case for the 173 stations under study. Gaugings are then ranked in chronological order. Consecutive gaugings in chronological order are hereafter called a gauging sequence or GS. Initial GS therefore included all available gaugings at a station. Relative residuals (RR) between discharges estimated from the adjusted baseline RC and gauged discharges are first estimated (Figure 2):

$$RR_j = \frac{Q_{RC}(h_j; a, b, c) - Q_j}{Q_j} \tag{4}$$

where $Q_j$ and $Q_{RC}(h_j; a, b, c)$ are the measured and estimated discharges for a given gauging. It is then assumed that changes in hydraulic conditions will manifest through a breakpoint in temporal RR series and Pettitt test (Pettitt 1979) was applied to detect these breakpoints (95% confidence level). Two additional conditions were considered. Firstly, the test was applied only to gauging sequence with 12 or more gaugings, and, secondly, breakpoint should not be located within the first or last six gaugings of a GS.





The GS is partitioned into two GS if a breakpoint is detected, and the two previous conditions are satisfied. The first GS
160     encompasses all gaugings before the date of occurrence of the breakpoint, and the second one all gaugings after the
breakpoint. Since Pettitt test can only detect one breaking point in a GS, the procedure is applied again to the newly
partitioned GSs until no further breakpoint point is detected or one of the previous conditions is not fulfilled (Figure 2).

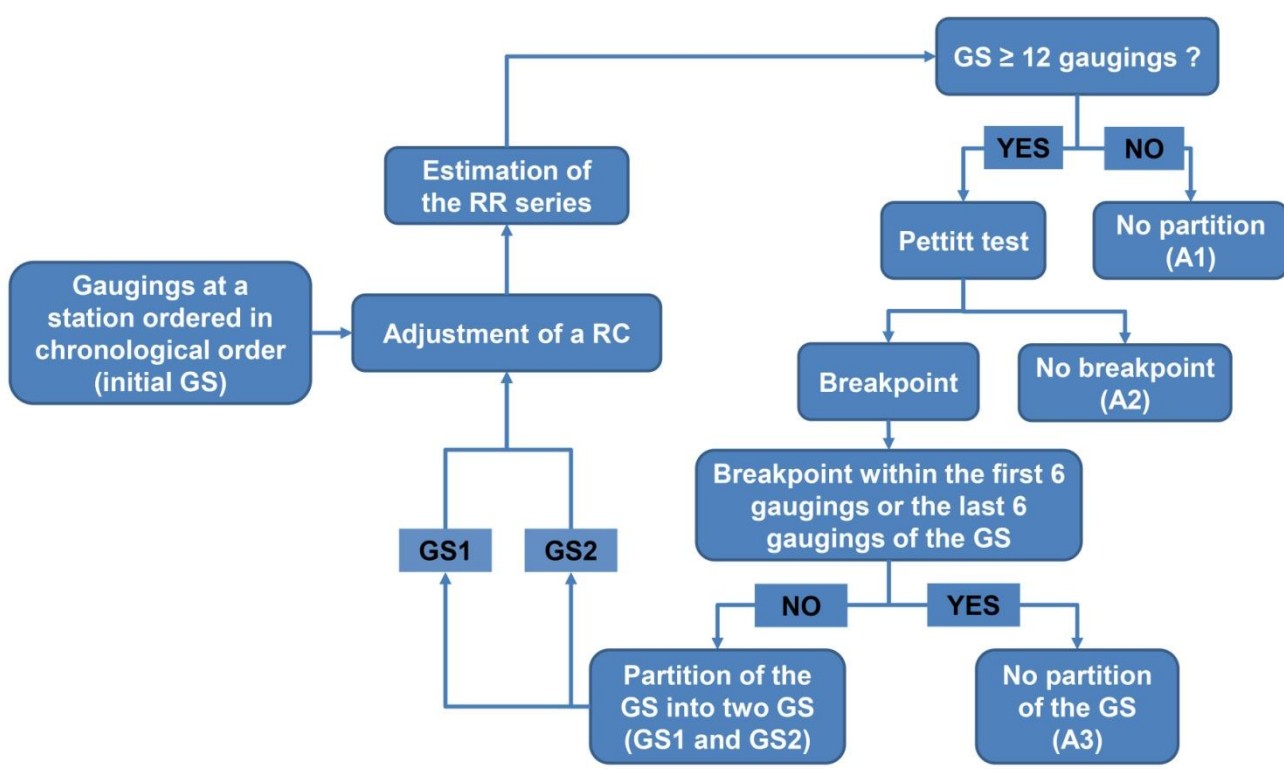

165

**Figure 2: Procedure to assess temporal changes in RC and partitioning of the initial gaugings sequences.**

Gauging sequences obtained after applying this procedure were classified into three groups (Figure 2): A1) GS with less than
12 gaugings (no breakpoint test applied); A2) GS with more than 12 gaugings and no breakpoint; A3) GS with more than 12
gaugings and a breakpoint detected within the first or last six gaugings. Group A3 was defined since it suggests that caution
is needed when adjusting RC to these GS, and if new gaugings are added. The total number of GS finally obtained at a
station corresponds to the number of periods and RC necessary to represent the stage-discharge relationship over the gauging
period at this station.



This procedure was applied to the 173 stations under study. As a result, 96 out of the 173 stations (55.5%) have one RC (no
change in RC over time), 31 stations (18.0%) two RC, while the remaining 46 stations have between 3 to 8 RC. The total
number of RC is therefore 348 for an average of 2.0 RC/station, and a Root Mean Square Relative Error (RMSRE) value of
7.3% with a 5-95% confidence interval ranging from 2.3 to 18.7%. The three GS with more than 12 gaugings and a
breakpoint detected within the first or last six gaugings (group A3) are not considered in the following.


It should be noted that the proposed procedure cannot account for changes in specific parts of the RC, for instance affecting
only the low flow part of the stage-discharge relationship. It can neither account for shift in RC due to progressive riverbed
changes, sedimentation, or erosion. Morlot *et al*. (2014) proposed a method to dynamically upgrade RC and associated
uncertainties. Jalbert *et al*. (2011) also proposed an approach based on variographic analysis to estimate the temporal
evolution of RC and associated uncertainties.

A similar approach was proposed by Darienzo *et al.* (2021). These authors use a more complex approach based on multi-
change point Bayesian estimation and applied the proposed approach to a station on the Ardèche river (France). Although, in
principle, very similar to the approach previously described, the complexity of Darienzo *et al*.'s approach limits its large-
scale application.

### 3.3 Adjustment of power functions to gauging sequences

Once the GS representing the stage-discharge relationship over a specific period have been identified, one must decide if one
or two power functions (PF) are needed to adequately represent the stage-discharge relationship over the entire range of
gauged water levels.

Figure 3 presents the general approach used in this study to determine the number of PF needed to represent the RC (only
one or two PF are considered). It is based on the hypothesis that residuals should be randomly distributed around the RC if it
adequately represents the stage-discharge relationship over the whole range of gauged water level, otherwise two PF are
needed. Randomness of residuals was assessed by applying a run test, also called randomness test, to the RR (Eq. 2) sorted
in increasing order of gauged water levels. The Wald-Wolfowitz run test (Wald and Wolfowitz 1940) was used (95%
confidence level). Randomness is tested only if the GS has 12 gaugings or more, otherwise it is assumed random, and one PF
is used to represent the RC. If RR series are not random according to the test, the transition level h', as well as the parameters
of the two PF are estimated using Eq. 3. If the estimated transition level is within the six smallest or the six largest gauged
water levels, then the corresponding GS is put aside for further analysis. Otherwise, the representation by two PF is kept and
the run test finally applied to the resulting RR series. No further PF is considered even if the run test concludes that the RR
of 2-PF RC are not random (Figure 3).





After application of the procedure presented in Fig. 3, RC are classified into the following groups: B1) RC represented by
one PF (random RR); B2) RC represented by one PF with transition level within the six smallest or the six largest water
levels; B3) RC represented by two PF (random RR); B4) GS possibly represented by more than two PF (non-random RR).
RC of group B4 should eventually be further investigated.

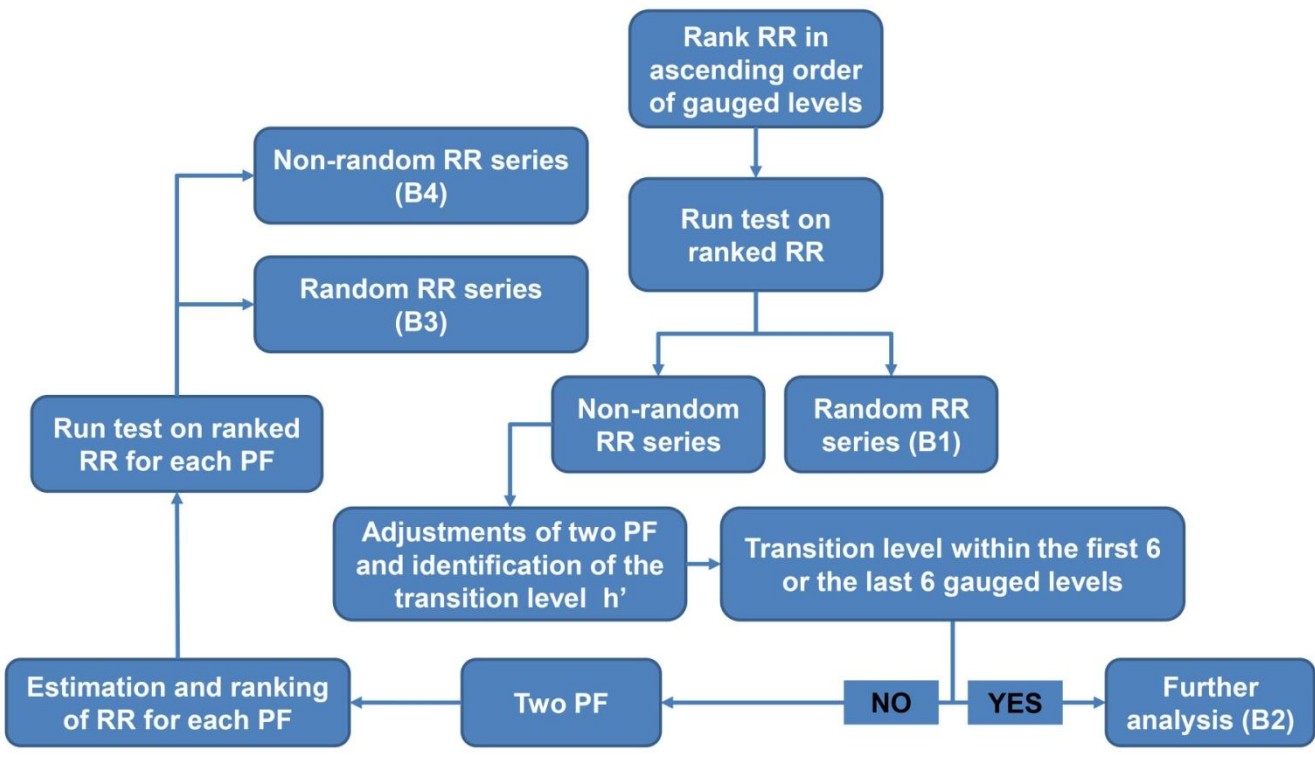


**Figure 3: Procedure to assess if one or two power functions (PF) are needed to represent the stage-discharge relationship of a given GS. The procedure is applied to all GS with 12 gaugings or more.**

The RC of the 55 GS with less than 12 gaugings are represented by one PF (no application of the run test). After applying
the procedure presented in Fig. 4, the remaining 290 GS are classified as follows: 245 in group B1 (one PF); 9 in group B2;
32 in group B3 (2 PF); 4 in group B4 (more than 2 PF). The four RC of group B4 and the 9 of group B2 were excluded from
the following analysis.



## 4 Estimated Rating Curves

A total of 332 RC was estimated, 300 (90.4%) represented by one PF, and 32 (9.6%) by 2 PF. Distributions of estimated

values of the exponent c of the fitted PF (Eqs 1 and 3) are shown on Fig. 4. As previously mentioned, these values can be related to the type of hydraulic control (Le Coz *et al*. 2014). The mean exponent of RC represented by 1 PF is close to 2, a value in-between those associated to rectangular and triangular control sections. Also, distributions of $c_1$ (low level PF) and $c_2$ (high level PF) for RC represented by 2 PF are very different. Larger $c_1$ values may be related to triangular-like sections, while smaller $c_2$ values to rectangular-like sections. Values well above $c = 8/3$ are also reported, especially for the $c_1$

exponent. Le Coz *et al*. (2014) mentioned that such values should be considered "suspicious". Hrafnkelsson *et al*. (2022) also proposed that exponent value should be within the range [1.0, 2.67]. In our case 18.0 % of the RC represented by 1 PF have exponent larger than 8/3, and 53.1% of those represented by 2 PF. Further investigation of these cases is required to better understand the hydraulic conditions leading to such values. Compensation effect between the three parameters when calibrating Eq. 1 may also explain these values. However, values larger than 8/3 cannot be *a priori* discarded.


The mean RMSRE value for the 300 1-PF RC is 7.1% and 5.7% for the 32 2-PF RC. A total of 141 RC (42.5%) displays RMSRE values less than 5% while only 52 RC (15.7%) have RMSRE values larger than 10%.

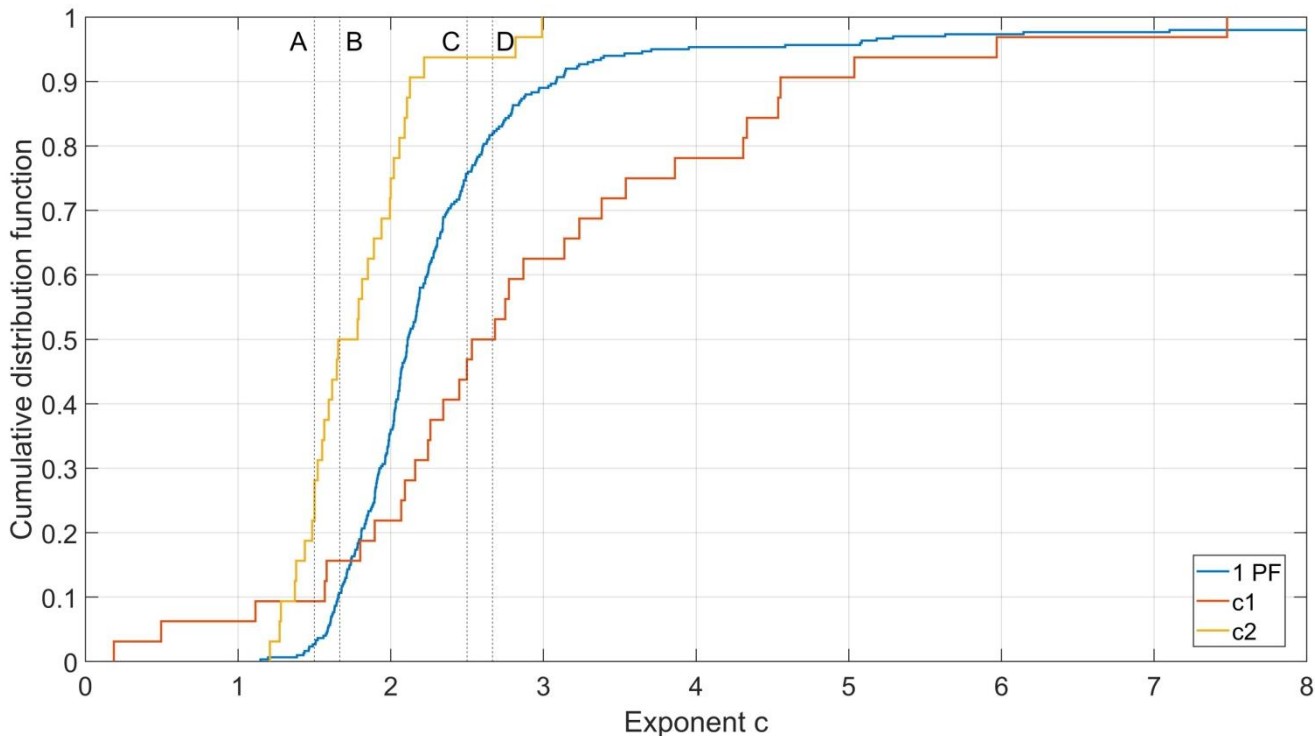

**Figure 4: Cumulative distributions of exponent c (Eq. 1) for the 300 RC represented by one PF, and the 32 RC represented by two**
**PF (Eq. 3) where $c_1$ corresponds to lower water levels and $c_2$ to higher water levels. Vertical dashed lines correspond to exponent**



of stage-discharge relationships for: A) rectangular weirs (c = 3/2); B) shallow permanent uniform flow in a rectangular canal (c = 5/3); C) triangular weirs (c = 5/2); D) shallow permanent uniform flow in a triangular canal (c = 8/3). The x-axis has been truncated at c = 8 for clarity.

## 5 Rating curve uncertainty models

Uncertainties on estimated RC were first investigated by looking at the RR distribution as a function of normalized discharge, defined as the discharges divided by the mean discharge of each RC, when the 300 1-PF RC are considered (Figure 5). Coxon *et al*. (2015) performed a similar analysis on 26 stable gauging stations in England and concluded that the RR were adequately represented by a logistic distribution with a discharge dependent standard deviation. Figure 5 shows that similar results were obtained for the 300 1-PF RC. Larger uncertainties for high and especially low flows are observed as

well as a slight bias for the highest discharges meaning that discharges estimated from the RC more likely overestimated gauged values.

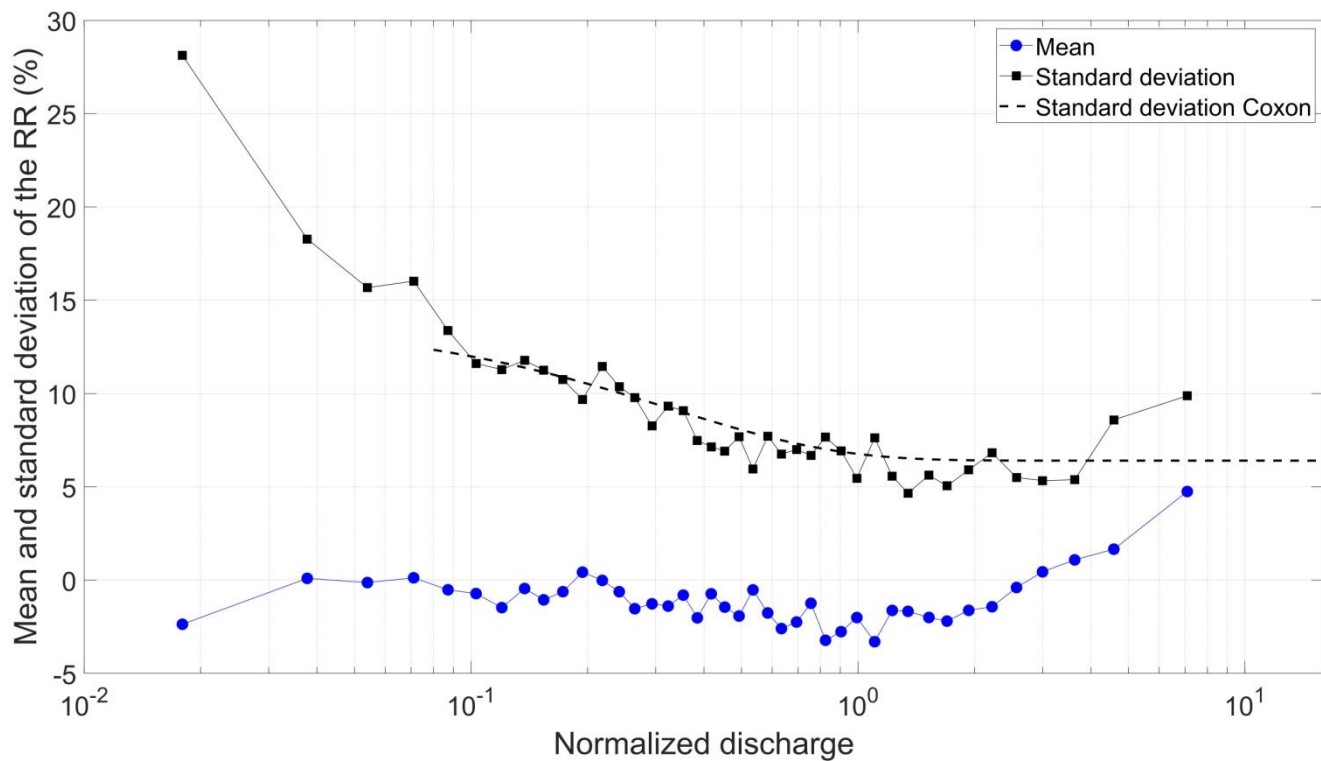

**Figure 5: Mean (blue) and standard deviation (black) of the RR of the 300 1-PF RC as a function of the mean normalized**
**discharge (each discharge interval includes ≈ 200 values). Dashed curve corresponds to the equation for the standard deviation of the logistic distribution obtained by Coxon *et al*. (2015). The range of normalized discharges covered by the dashed curve corresponds to the one of Fig. 4 in Coxon *et al*. (2015). Note the logarithmic x-axis.**





L-moment diagram (Hosking and Wallis 1997) of RR distribution indicates that distribution skewness is close to zero

(symmetrical), while kurtosis is much larger than the kurtosis of normal distribution and even slightly larger than the kurtosis of logistic distribution (not showed for conciseness) therefore suggesting that the logistic distribution more adequately represents the RR empirical distribution as in Coxon et al. (2015) (see Fig. 5). Finally, the standard deviation of the RR for normalized discharge between $\approx 0.25$ and $\approx 5$ are smaller than 10%, with a minimum value of 5%, values comparable to reported uncertainties on flow measurements (McMillan et al. 2012).

Uncertainty models for the rating curves were therefore developed based on the previous analysis. Relative normalized stage, $h'$, is first defined:

$$h' = \frac{(h - h_{min})}{(h_{max} - h_{min})} \tag{5}$$

where $h_{min}$ and $h_{max}$ correspond respectively to the Smallest Gauged Stage (SGS) and to the Largest Gauged Stage (LGS) at a station, and we therefore have $0 < h' < 1$.

Normal and logistic distributions were considered for uncertainties on estimated discharges from RC. Location parameters are set to zero and four models are considered for the scale parameter $\sigma(h')$, based on the results of Figure 5:

$$\sigma_{M0}(h') = \sigma_0 \tag{6a}$$

$$\sigma_{M1d}(h') = \sigma_1 \exp(-\alpha_1 h') + \sigma_0 \tag{6b}$$

$$\sigma_{M1i}(h') = \sigma_2 \exp(\alpha_2 h') + \sigma_0 \tag{6c}$$

$$\sigma_{M2}(h') = \sigma_1 \exp(-\alpha_1 h') + \sigma_2 \exp(\alpha_2 h') + \sigma_0 \tag{6d}$$

with parameters $\sigma_m$, $m = \{0,1,2\}$ and $\alpha_k$, $k = \{1,2\}$. Model M0 corresponds to a homoscedastic uncertainty model, while models M1d, M1i, and M2 correspond to heteroscedastic models. All parameters are positive, and the condition $\sigma_0 > 2\%$ was set to avoid unrealistic standard deviations. Figure 6 presents the relative uncertainties as a function of stage according to these models.





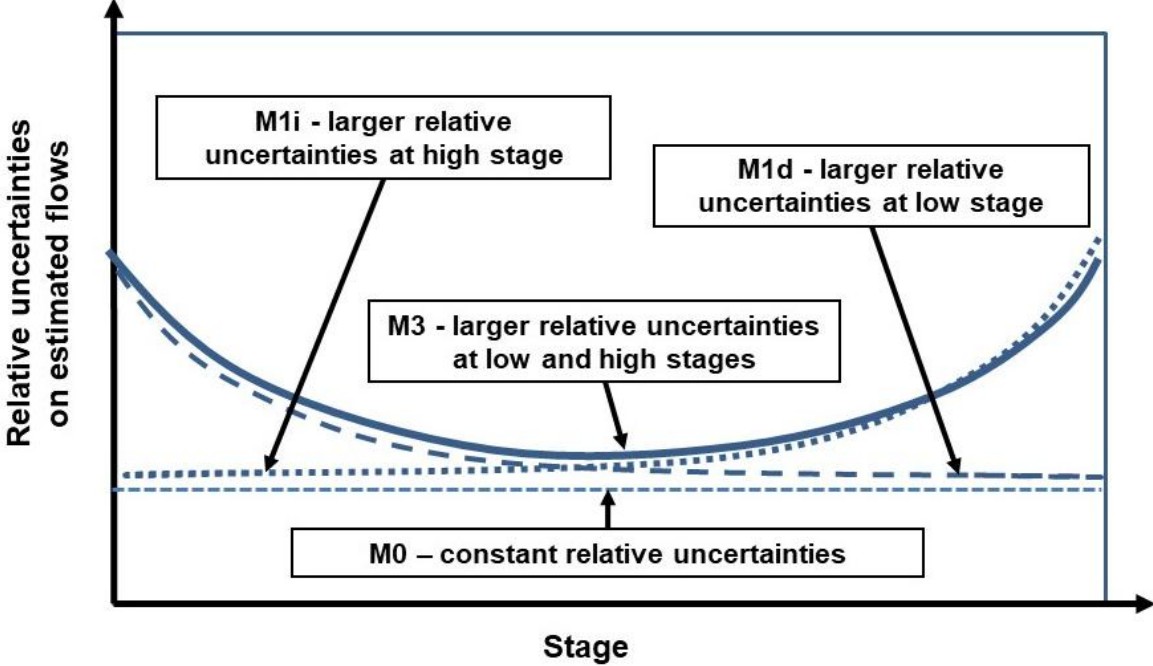


**Figure 6: Relative uncertainties on estimated flows as a function of stage for the various uncertainty models: M0 - constant (homoscedastic model; Eq. 6a); M1d - decreasing (Eq. 6b); M1i - increasing (Eq. 6c); M2 - decreasing for small stages and increasing for large stages (Eq. 6d).**

The various uncertainty models have been constructed by combining the normal (N) and logistic (L) distributions to the four

models describing the stage dependency of the scale parameters (M0, M1d, M1i, M2). A total of eight models was therefore

considered: 1) N-M0; 2) N-M1d; 3) N-M1i; 4) N-M2; 5) L-M0; 6) L-M1d; 7) L-M1i; 8) L-M2. Therefore N-M0 model

corresponds to the case where uncertainties on estimated flow are represented by a normal distribution with constant scale

parameter (relative uncertainties independent of stage), while L-M1d corresponds to the case of uncertainties on estimated

flow are represented by a logistic distribution with decreasing scale parameter as stage increases (larger relative uncertainties

at low stages). Parameters of each uncertainty models were estimated by maximizing the corresponding likelihood functions.

Results from the different models were then compared using the Bayesian information criterion (BIC; Burnham and

Anderson 2020). The model minimizing the BIC was selected.

Table 1 presents a summary of the results for the 296 RC with 10 gaugings or more. The normal distribution is the most

frequently selected distribution (67.2%). Homoscedastic model is selected for 75.0% of the RC. Models L-M1i, N-M2 and

L-M2 are never selected. Consequently, the dominant uncertainty models are the homoscedastic models, N-M0 (47.6%)

followed by L-M0 (27.4%), while the heteroscedastic models, N-M1d and L-M1d, both assuming decreasing relative

uncertainties as stage increases, are selected for 24.7% of the RC (Figure 6). These results suggest that the adequacy of the





RC to represent the small discharges can be problematic in many cases. Model with increasing uncertainties with stage, N-
Mi, is selected one time (this case is not considered in the following).

**Table 1. Number of RC (percentages) and mean number of gauging per RC for each selected uncertainty models. Note that the L-M1i, N-M2 and L-M2 models are never selected. Only RC with 10 or more gaugings are considered.**

| Uncertainty model | Number of RC (%) | Mean number of gauging per RC |
|---|---|---|
| Normal distribution with constant scale parameter (N-M0) | 141 (47.6) | 24.3 |
| Normal distribution with decreasing scale parameter as stage increases (N-M1d) | 57 (19.3) | 45.4 |
| Normal distribution with increasing scale parameter as stage increases (N-M1i) | 1 (0.3) | 54.0 |
| TOTAL N distribution | 199 (67.2) | 30.5 |
| Logistic distribution with constant scale parameter (L-M0) | 81(27.4) | 29.5 |
| Normal distribution with decreasing scale parameter as stage increases (L-M1d) | 16 (5.4) | 45.2 |
| TOTAL L distribution | 97 (32.9) | 32.1 |






A look at the distributions of the number of gaugings per RC (not shown for conciseness) and at the mean number of
gaugings per RC (Table 1) shows that homoscedastic models (M0) are more often selected when the mean number of
gaugings is smaller, while heteroscedastic models (M1d and M1i) are more often selected when the mean number of
gaugings is larger. This is consistent with the hypothesis that RC based on a larger number of gaugings will more likely
explore broader hydraulic conditions, and therefore more adequately represent the discharge-stage relationship for both low
and high flow regimes where uncertainties may increase due to a change in control. Figure 7 presents two examples of RC
where the L-M0 and N-M1d models were selected.

Figure 8 presents the distributions of standard deviations for RC with the M0 and M1 uncertainty models. Mean standard
deviation for N-M0 models is 6.2% with a maximum value of 25.5%, while mean standard deviation for L-M0 models is
7.0% with a maximum value of 24.5%. Therefore, 86.4% of the RC with homoscedastic uncertainty models have
uncertainties smaller than 10%. For the RC with M1 heteroscedastic uncertainty models (N-M1d, N-M1i, and L-M1d),
uncertainties are smaller than 10% for a large majority of water levels within the interval $0.1 < h' < 0.8$. Uncertainties
increase significantly for high, and especially low stages, and can be larger than 20% in some cases. This shows that caution
is needed when estimating flow corresponding to the low and high range of the gauged levels as uncertainties may increase.





**Figure 7: RC with 2.5-97.5% confidence intervals at: a) Mitchinamecus - 040619 (45 gaugings from 1977 to 2019; L-M0 model with standard deviation of 4.0%); b) Matapedia - 011509 (60 gaugings from 1996 to 2022; N-M1d model). Note the y- and x-axis logarithmic scales. Stage is expressed as the difference between the level and the reference level (b in Eq. 1).**



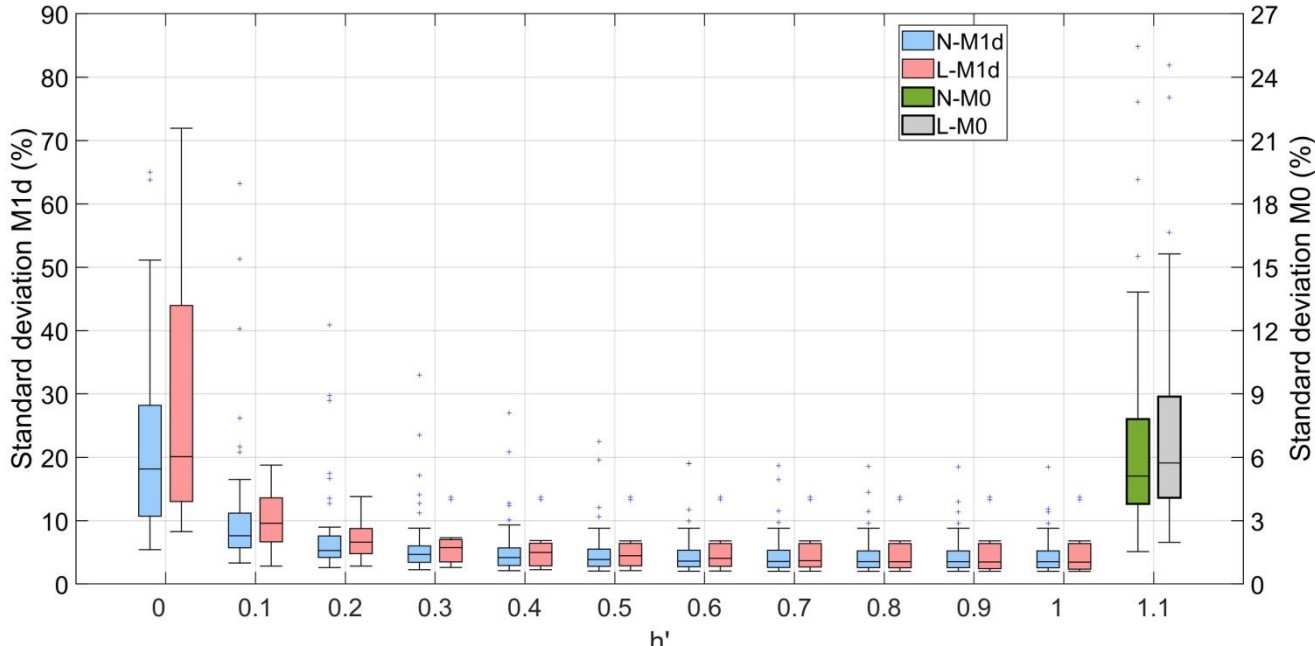

**Figure 8: Box plots of the standard deviations for the 57 RC with N-M1d (blue), the 16 RC with L-M1d (pink) uncertainty models**
**a function of normalized level h' (Eq. 5). Box plots for the 141 RC with N-M0 (green) and 81 RC with L-M0 (grey) models are displayed on the right-hand side of the graph and refer to the right y-axis. The upper limit of the left y-axis was set to 90% for convenience.**

## 6 Summary and conclusions

Stage-discharge relationships at specific control section or reach are represented by rating curves (RC). Hydrometric stations
adequately located at these sites record water levels that can be used to estimate discharge using these RC. These data are crucial input for many practitioners responsible of various water management projects such as flood mapping or water supply. Providing uncertainties on estimated discharges from RC is therefore essential.

Establishing a RC at strategic sites requires that measured discharge and corresponding water levels be monitored over a
long period to cover diverse flow regimes. Mathematical functions are then adjusted to these gaugings and use to 'interpolate' or 'extrapolate' the stage-discharge relationship to ungauged water levels. Power functions (PF) are usually used but other mathematical functions, such as the splines, can be used as well. PF often appears in formulas describing stage-discharge relationship of standard hydraulic structures such as weirs. The exponent of these PF, called hydraulic exponent, can therefore be related to the type of hydraulic controls.


Many factors must be accounted for when developing a RC, among which possible irreversible abrupt or long-term changes in stage-discharge relationship due to modifications in watercourse geomorphology, hydraulic controls, or sedimentation. As





a result, a RC developed from past gaugings may be inadequate to represent forthcoming stage-discharge relationship. New gaugings must be realized to update the RC. Another important factor is the adequacy of a single PF to represent the stage-discharge relationship over the whole range of gauged water levels. In some instances, two PF or more may be necessary to account for possible changes in hydraulic controls.

Many approaches have been proposed in the literature to deal with these issues, but their relative complexity prevents a large-scale application. This paper proposes a simple approach to develop RC and estimate corresponding uncertainties that can be readily applied to hydrometric networks. It accounts for possible temporal changes in RC over time and determines if one or two PF are required to represent the gauged stage-discharge relationship. Uncertainty models for estimated flows from RC are selected through the analysis of relative residuals of adjusted RC.

The approach was applied to the Quebec hydrometric network, which includes 173 hydrometric stations mainly located in southern Québec. The number of gaugings at these stations ranges from 9 to 197 (mean value of 58.3) and covers periods from 3 to 98 years (37.7 years in average). Temporal partition of initial gauging sequences was first performed to identify sequence of consecutive gaugings representative of the RC over specific period. Single RC can be used to represent the available gauging period at 96 (55%) stations of the Québec hydrometric network, while stage-discharge relationship at 31 (18%) stations must be divided into two periods and therefore represented by 2 RC. The remaining gauging periods at the 46 stations are subdivided into 3 to 8 periods with corresponding number of RC. Three RC were set aside following this analysis. The total number of RC is therefore 348, for an average of 2.0 RC/station, and 29.0 gaugings/RC.

Adjustment of PF to these 348 RC was then carried out. Sixteen RC were discarded (more than 2 PF or transition level in the six lowest or six highest gauged levels). Of the remaining 332 RC, 300 (90.4%) are represented by one PF, and 32 (9.6%) by 2 PF. The hydraulic exponents for RC represented by a single PF range from 1.1 to 18.8 with a mean value of 2.5. Most of these values are within the range of standard hydraulic controls (e.g., rectangular, or triangular weirs, uniform flow). Hydraulic exponents for RC represented by 2 PF are different with larger exponents for the low stage part of the RC, and smaller exponents for the high stage part of RC. These values may be associated to most likely triangular-shaped sections at low stage (larger exponents), and most likely rectangular-shaped sections at high stage. Values well above $c = 8/3$ (associated to triangular-shaped weir) are also reported for 18.0% of the RC represented by 1 PF, and 53.1% of those represented by 2 PF. Further investigation of these cases is required to better understand the hydraulic controls at these sites and the reasons such large values are obtained.

Relative residuals (RR) between the gauged discharges and those estimated from the RC have been used to identify the appropriate uncertainty model describing RR distribution of each RC. The normal (N) and logistic (L) distributions were used and four models were considered to represent the scale parameter dependency with stage, a first one with constant scale





parameter (homoscedastic; M0), and three others with stage-dependent scale parameter (heteroscedastic), associated to decreasing (M1d), increasing (M1i), and U-shaped (M2) scale parameter as stage increases. Combining the two distributions with the four scale parameter models result in eight uncertainty models ({N, L} x {M0, M1d, M1i, M2}).


The uncertainty model that best fit the RR distributions was identified for each RC. Homoscedastic models are predominantly selected (47.6% for the L-M0 and 27.4% for the N-M0 model), while the preferred heteroscedastic models are the N-M1d (19.3%) and the L-M1d (5.4%) both associated to larger relative uncertainties at low stage. Therefore, adequacy of the RC to represent small discharges could be problematic for many RC, a situation that may be explained by

possible change in hydraulic controls at low flow regime. Only one RC have increasing relative uncertainties with stage while U-shaped uncertainty model is never selected. This may be indicative that high stages are, in many cases, inadequately covered by gauging campaigns, which is not surprising considering the risks and challenges of gauging in such conditions.

Standard deviations for the RC represented by given uncertainty models were estimated. Mean standard deviation for N-M0

models is 6.2% with a largest value of 25.5%, while mean standard deviation for L-M0 models is 7.0% with a largest value of 24.5% respectively. Therefore, the mean uncertainties of RC represented by homoscedastic models is 6.5%. For RC with heteroscedastic models, uncertainties for mid-range gauged stages are comparable to those of homoscedastic models but increase significantly for low stages, and can reach values larger than 20%.

Caution is recommended when using RC to estimate flow corresponding to the low or high range of the gauging levels used to construct the RC as uncertainties may rapidly increase and be inadequately represented by the selected uncertainty models. This could be even more problematic if flows are estimated from the extrapolated part of the RC, i.e., below the smallest or above the largest gauged stage. Gaugings may not, or incompletely, cover flow conditions where hydraulic control may change and not be well-represented by available gaugings and resulting RC. This could result in bias in

estimated discharges, and estimated uncertainties based on the interpolated part of the RC may be misleading. This is an important issue for the statistical analysis of flood frequency since the maximum gauged stage are smaller than the 5-year annual maximum measured water level for 52% of the stations and smaller than the 2-year annual maximum measured water level for 31% of the stations (this analysis was not presented for conciseness). Estimated annual maximum flow used in frequency analysis can therefore be highly biased and uncertain 'Representativity' of the range of available gaugings should

be assessed to provide guidelines to practitioners responsible of the hydrometric stations for future gaugings campaign, and data users.

The following points should be further investigated. The proposed procedure to assess the 'stationarity' of RC assumed that, once it is stated that a RC needs to be updated, all past gaugings are discarded and that forthcoming gaugings are used to

construct the 'new' RC. An alternative approach would be to identify if parts of RC are still relevant and should be kept when updating the RC such as in the method proposed by Morlot *et al*. (2014).

As stated previously, hydraulic exponents of the power function may be linked to the type of hydraulic control in play. Estimated hydraulic exponents obtained after this large-scale application to Quebec hydrometric network are, for the most

part, within the range of values of usual hydraulic structures (weirs, gauging flumes, etc.). However, large values have been reported at some stations. These sites should be further investigated, and the broader issue of the added value of extensive hydraulic analysis (Di Baldassarre and Montanari 2009; Lang *et al*. 2010; Mansanarez *et al.*, 2019b), or quantitative assessment of the hydraulic controls (such as in the BaRatin method proposed by Le Coz *et al*. 2014) should be also further investigated.

**Author contribution**

CF and AM managed the project. CF and her team at the MELCCFP provided the gauging measurements data. AM developed the methodology. AM and GT analyzed the data. AM and SB conducted the bibliographical research. AM wrote the manuscript draft. GT, SB and CF reviewed and edited the manuscript.

**Competing interests**

The authors declare that they have no conflict of interest.

**Acknowledgements**

The authors would like to thank the INFO-Crue program for financing this project. They also thank the following persons for their insightful thoughts: Alexandrine Bisaillon, Mohammad Bizhanimanzar, and Gabriel Rondeau-Genesse from Ouranos, Thomas-Charles Fortier-Fillion and Richard Turcotte from the *Ministère de l'Environnement, de la Lutte contre les*

*Changements Climatiques, de la Faune et des Parcs*. The authors would also like to acknowledge the contributions and suggestions from the members of the project advisory committee: Thibault Mathevet (*Electricité de France*), Stéphanie Moore (Environment and Climate Change Canada) and Elaine Robichaud (Hydro-Québec).

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
