# Peer review of "Assessment of uncertainties on stage-discharge rating curves: A large scale application to Québec hydrometric network"

_EGUsphere, 2024_

## Author Response (AR1)

**Assessment of uncertainties on stage-discharge raging curves: A large scale application to Québec hydrometric network**

**Alain Mailhot, Guillaume Talbot, Samuel Bolduc, Claudine Fortier**

**Point-by-point reply to editor and reviewers' comments**

A point-by point response to editor and reviewer's comments is provided in the following with corresponding references to the revised version of the manuscript. Since the numbering of the lines changed, we indicate in each case the new line numbers where revised text appears. Please note that the changes in the 'marked' version are indicated in green.

**Editor report**

**Comment 1** After reading the reviewers comments, we can conclude that the manuscript needs extensive proof reading. One of the values of this analysis relies on considering a very large number of gauging stations covering an extensive hydrometric network located in South Quebec.

Following the comments from the first reviewer, it would be beneficial to better clarify the goal and focus on the uncertainty assessment method used in this research. The reviewer seems to approach uncertainty estimation from a "Bayesian" point of view; this is, by considering unknown parameters as well as the predictions as random variables. A clearer statement on the uncertainty assessment method used in this research might clarify this point, and how it differs from the reviewer's proposal.

**Response** We add some clarifications about the uncertainty assessment method as suggested by the editor and the reviewers. A paragraph has been added at the end of the introduction providing an overview of the proposed approach (lines 100-108).

**Comment 2** Including the proposed discussion about the c parameter in equation (1) would be very useful, as well as the proposed changes related to the manuscript grammar, etc.

**Response** We add a paragraph discussing the 'unusual' c values (lines 233-249). See also our response to comment 4 from reviewer 1. The manuscript has been carefully revised and all corrections proposed by the reviewers were made.

**Comment 3** Limitations of this approach as suggested by reviewer two, should also be addressed.

**Response** Reviewer's comment about the overrepresentation of medium stage levels has been addressed on our response (see our response to Comment 2 from Reviewer 2). We also add a sentence in the conclusion of the paper about this issue (lines 414-415).

**Reviewer #1**

**General comments**

This paper presents a robust method to fit rating curves to hydrometric networks, that is, to fit a large number of rating curves over a long period in an automatic manner. The method accounts for possible temporal changes in RCs over the observed period of each station and determines whether one or two power-law rating curves are required to represent the gauged stage-discharge relationship. The paper also proposes models for representing a stage-dependent scale parameter. The proposed method is applied to the Québec hydrometric network, which includes 173 hydrometric stations mainly located in southern Québec.

**Specific comments**

**Comment # 1** The uncertainty of model parameters is not addressed in this paper. Thus, the uncertainty in the prediction of discharge for a particular value of the stage, h, corresponding to the rating curve only, can't be accessed. This is true for both stage h within the observed stage-discharge pairs and stage h outside the range of the observed stage-discharge pairs. If the uncertainty of future discharge observation is needed, then both the uncertainty of the model parameters and the uncertainty due to the error term (representing the measurement error and potential model error) are needed. The uncertainty of the error terms is addressed in this paper but that alone is not enough to address uncertainty in predictions. Please evaluate the uncertainty of the model parameters and report this uncertainty.

**Response**: In our approach, the uncertainty models are built from the residuals, i.e. the difference between the observed and estimated values. Various uncertainty models are considered (e.g. homoscedastic and heteroscedastic) and adjusted to the distribution of residuals. We therefore assume that all sources and types of uncertainties (stage, flow, model parameters, rating curve uncertainties) are accounted for in residual distributions. It is a straightforward approach where there is no need to identify or separate uncertainty contributions. By doing so we assume that residual distributions are representative of uncertainties on future discharge estimated from the rating curve. This would be inappropriate if some source of uncertainties, for example, more precise flow measuring devices were used at some point in time. Such information is not considered in our approach. We assume that uncertainties remain unchanged over the process of the construction the rating curve.

We therefore disagree with the reviewer when he says that, since our approach didn't specifically address model parameter, it cannot be used to address uncertainty in predictions. We think that, even though this approach doesn't explicitly identify and set a statistical framework of the various uncertainty contributions to total uncertainties, it is rigorous and useful.

A paragraph has been added at the end of the introduction providing some clarifications about the proposed approach (lines 100-108).

**Comment # 2** Following Comment 1), it should be described how the uncertainty of the model parameters enters predictions of discharges for h above the range of the observed stage-discharge pairs. Furthermore, the prediction performance of the method should evaluated by leaving out 2-4 observed stage-discharge pairs holding the largest stage values. As the left-out observed data pairs include measurement error then the prediction should take into account the uncertainty in both the model parameters and the error term.

**Response**: The proposed approach doesn't specifically address the uncertainties below the Smallest Gauged Stage (SGS) or above the Largest Gauged Stage (LGS). In fact, the uncertainty models estimated in the interpolated part are used to assess uncertainties in extrapolated parts of the rating curve, and cautionary remarks are made about this issue in the paper. For instance, Figure 8 shows that uncertainties for many rating curves increases as stage decreases which can be reasonably related to a progressive change of hydraulic control. Uncertainty models can partly account for such changes but a more detailed analysis of the hydraulic is needed to further assess uncertainties for these parts (below the SGS and above the LGS) of the rating curve. Leaving out 2 or 4 gauged stage-discharge pairs with the largest stages, as suggested by the reviewer, may be viewed as a possible way to estimate the uncertainties in the extrapolated part of the RC, but remains a questionable approach considering that the main source of uncertainties is related to changes in hydraulic control or to changes in channel characteristics. Therefore, we didn't use resampling approaches and preferred to stress the limits of our approach (and of any resampling approaches) and to reiterate the importance of hydraulic analysis to investigate uncertainties above the LGS and below the SGS.

**Comment # 3** Please provide a reference for the function in Eq. (2) and explain what it is based on. Is it based on the likelihood function of a particular stochastic model? There should be a discussion about other modeling approaches and other inference schemes for discharge rating curves found in the literature. The approach based Eq. (2) only focuses on finding point estimates for the unknown parameters but there is no mention of uncertainty in the parameters due to the lack of information in the data (as mentioned in Comment 1).

**Response**: Mean Square Relative Error (MSRE) is a normalized measure of the difference between estimated and observed values often used in hydrology (see for example Gupta *et al*. 2009). It is based on the hypothesis that the best adjustment between model and observed values (flows estimated by the power function and gauged flows) is obtained through the minimization of this metric. Error models are developed from the distribution of this discrepancy measure. Intuitively, we can expect that an adequate RC should correspond to the one where relative errors don't depend on stage (homoscedastic model). Results support this hypothesis in most of the cases since 75% of the RC are adequately represented by a normal or logistic homoscedastic distribution with mean standard deviation of the order of 5-7%, values close to the estimated flow measurements. However, stage dependent-error model is retained for 25 % of the RC.

Considering that MSRE is very commonly used we didn't think it would be useful to add a reference for function in eq. 2.

Gupta, H.V., Kling, H. Yilmaz, K.K., Martinez, G.F. (2009). Decomposition of the mean squared error and NSE performance criteria: Implications for improving hydrological modelling, *J. Hydrol.* 377 (1-2): 80-91. DOI: 10.1016/j.jhydrol.2009.08.003.

**Comment # 4** In Section 4, it is mentioned that values of c larger than 8/3 should not be *a priori* discarded. This is a reasonable claim for values of c close to 8/3, however, values of c greater than 5 are questionable for natural open channels. For example, the inverse parabolic cross section has c between 3.67 and 4.33 (Hrafnkelsson *et al.*, 2022), and it is unlikely to appear in nature. Values of c greater than 5 correspond to cross sections more extreme than the inverse parabolic cross section, that is, narrow width close to the bottom and a large increase in width with the stage moving further from the bottom. Estimates of c greater than 5 are more likely to be observed due to sampling error than extreme cross sections. Furthermore, estimates of c that are less than 1 are most likely observed due to sampling error. Please consider tackling this issue by adding a penalty term to Eq. (2) (or to a function other than Eq. (2)) that penalizes values of c that are too large (c > 5) and too small (c < 1). This can, for example, be approached by using a parabolic term resembling the logarithmic transformation of a Gaussian prior density. Please consider also adding a penalty term for the parameters b (in the 1 PF model) and b_1 (in the 2 PF model) that ensures that b and b_1 are less than the smallest measured water level and penalizes for too small values of b and b_1.

**Response**: Of the 361 values used to build Figure 4, three of the c exponents for 1 PF rating curve, and 14 of the c1 exponents for the 2 PF rating curve have values larger than 5, which represent less than 5% of the estimated c values. As explained in the paper, these peculiar cases were not investigated prior to the initial submission of the manuscript but were further analyzed since the submission of the paper. Our analysis examined all RC with c > 3 and concluded that they occur in four situations:

1. **Small number of stage-discharge pairs and small range of gauged stages**: Many unusual c values are observed under these conditions. It is incidentally the case for the three RC with c < 1. Also in some cases, even if the number of gauging may seem reasonable, the range of gauged flow and stage remains small. The available stage-discharge pairs 'under sampled' the stage range and, as suggested by the reviewer, may be explained by sampling errors.
2. **Hydrometric stations downstream of dams**: Streams downstream of dams often have much smaller flow variability and therefore much narrower range of gauged stage. The largest reported c values correspond to this situation.
3. **Control section upstream a natural spillway**: In some cases, hydraulic control corresponds to a natural spillway. This type of configuration is often observed but could be problematic if the crest of the natural spillway is close to water surface at low stage. In such case, the adjusted 'b' value of equation 2 is unrealistic, 'a' is very small, and must be compensated by a large 'c' value.

4. **Compensating effect between a and c parameters of equation 2**: In some cases, under specific conditions, e.g. when small stages are over-represented compared to high stages, adjusting the CT led to very small 'a' values and large 'c' values. This effect is often observed, but not only, downstream dams or natural spillways as mentioned previously.

As can be seen, these unusual 'c' values are related to very peculiar situations suggesting that power function may be tricky to estimate in these cases. We prefer to explain the reasons for such values instead of adding a penalty term, as suggested by the reviewer, that would 'artificially constrained' c values.

We therefore add a discussion like the previous one at the end of Section 4 that explains the various conditions leading to these unusual c values (lines 233-249).

**Technical corrections**

**Lines 9-10**: Please consider changing "usually represented by power functions, a mathematical function" to "usually represented by power functions, mathematical functions".

**Response:** Done (lines 10-11).

**Lines 9-10**: Please mention that a power function in the context of discharge rating curve models is also referred to as the power law in the literature.

**Response**: Done (line 11).

**Line 14**: Please consider changing "This paper proposed an approach" to "This paper proposes an approach".

**Response**: Done (line 15).

**Line 16**: Please consider changing "and assessed if one or two" to "and assesses whether one or two".

**Response**: Done (line 18).

**Line 22**: Please add a comma in "they increase significantly at low stages, reaching values larger than 20%".

**Response**: Done (line 24).

**Line 38**: Please change "bias on estimated flows" to "bias in estimated flows".

**Response**: Done (line 40).

**Lines 67-70**: Here the main representations of RC are reviewed. However, only the power-law representation and the approaches based on cubic splines and Chebyshev polynomials are mentioned while multi-segment discharge rating curves (e.g. Petersen-Øverleir, A., and Reitan, T., 2005; Reitan, T., and Petersen-Øverleir, A., 2009; Hodson *et al.* 2024) and the generalized power-law rating curve (Hrafnkelsson *et al.*, 2022) are not mentioned. Please consider revising the overview of RC.

**Response**: Overview of RC has been updated and references added (lines 65-68).

**Lines 73-74**: It is stated that „Despite these legitimate criticisms, wide application of the power function has shown that stage-discharge curves are surprisingly well-represented by such functions, which is the case in the actual application." It is true that the power function works well in actual applications but that is only true for a certain proportion of rivers as it has been shown that for some rivers the power function is inadequate (e.g., Petersen-Øverleir and Reitan, 2005; Reitan and Petersen-Øverleir, 2009; Hrafnkelsson *et al.,* 2022; Hrafnkelsson et al., 2023). Please consider rephrasing the above statement.

**Response**: The sentence has been changed accordingly and references added (lines 75-76).

**Lines 68-70**: After mentioning that more than one power function must be used it would be appropriate to cite papers on multi-segment rating curves, e.g., Petersen-Øverleir and Reitan (2005), Reitan and Petersen-Øverleir (2009), Hodson et al. (2024).

**Response**: References to theses papers have been added (lines 65-68 and 88).

**Lines 92-95**: The main objective of the paper is mentioned here, that is, modeling of error terms (or the residuals) of the models. Here previous work on models for the error terms should be cited (e.g., Reitan and Overleir 2004, Hrafnkelsson *et al.* 2012; Hrafnkelsson *et al.* 2022).

**Response**: A sentence mentioning the use of a similar approach and citing the relevant papers (lines 106-108). We didn't find reference Reitan and Overleir 2004 and assumed that the reviewer was referring to the following papers:

Reitan, T., and Petersen-Øverleir, A. (2006). Existence of the frequentistic estimate for power-law regression with a location parameter, with applications for making discharge rating curves. Stochastic Environmental Research and Risk Assessment, 20 (6), 445-453, https://doi.org/10.1007/s00477-006-0037-6.

**Line 116**: Section 3 might be better described with a title including the word "method" as opposed to the word "approach". Please consider revising the title of Section 3.

**Response**: Done (lines 129-130).

**Line 120**: Please consider changing "The following sections further details each of these steps. " to "The following sections provide further details for each of these steps. ".

**Response**: Done (line 132).

**Line 128-130**: Maybe it is sufficient to state that "Relative errors on estimated discharge were considered. However, this assumption will be further investigated. "?

**Response**: We agree, and the sentence has been changed accordingly (lines 138-139)

**Line 131**: Please explain why six gaugings are set as the minimum to adjust a single RC.

**Response**: Obviously, a minimum number of gaugings must be set to adjust a power law. The choice of six gaugings is subjective.

**Lines 135-139**: Please provide reference/references after "hereafter called the transition level. " in Line 139.

**Response**: We proposed this expression and therefore no reference can be provided.

**Line 177**: Please consider changing "an average of 2.0 RC/station" to "an average of 2.0 RC per station".

**Response**: Done (line 179 and line 375).

**Line 178**: Please consider changing "a 5-95% confidence interval" to "a 90% confidence interval".

**Response**: Done (line 180).

**Lines 177-178**: It is reported that the RMSRE value is 7.3% with a 90% confidence interval equal to (2.3%,18.7%). It would help to note that this estimate is based on the relative residuals from all the stations.

**Response**: The precision has been added (line 180).

**Line 189**: Please change "the complexity of Darienzo et al.'s approach" to "the complexity of Darienzo et al. (2021) approach".

**Response**: Done (line 191).

**Lines 254-257**: Adding 95% confidence intervals to the sample mean (blue dots) and sample standard deviation (black squares) of the RR would improve Figure 5.

**Response**: Figure 5 was changed as proposed by the reviewer (Figure 5 on page 12).

**Line 284**: In Figure 6, the box for model M2 uses M3 instead of M2. Please revise Figure 6 accordingly.

**Response:** The legend was corrected (Figure 6 on page 13).

**Lines 306-309**: The model L-M1d should be referred to as Logistic distribution but not as Normal distribution.

**Response**: The correction has been made (Table 1 on page 14).

**References**

Hodson, T. O., Doore, K. J., Kenney, T. A., Over, T. M., and Yeheyis, M. B. (2024). Ratingcurve: A python package for fitting streamflow rating curves. Hydrology, 11(2).

Hrafnkelsson, B., Ingimarsson, K.M., Gardarsson, S.M., Snorrason, A. (2012). Modeling discharge rating curves with Bayesian B-splines. Stoch. Env. Res. Risk A. 26 (1), 1-20.

Hrafnkelsson, B., Sigurdarson, H., Rögnvaldsson, S., Jansson, A. Ö., Vias, R. D., and Gardarsson, S. M. (2022). Generalization of the power-law rating curve using hydrodynamic theory and Bayesian hierarchical modeling. Environmetrics, 33(2):e2711

Hrafnkelsson, B., Vias, R. D., Rögnvaldsson, S., Jansson, A. Ö., and Gardarsson, S. M. (2023). Bayesian discharge rating curves based on the generalized power law. In Hrafnkelsson, B., editor, Statistical Modeling Using Bayesian Latent Gaussian Models : With Applications in Geophysics and Environmental Sciences, pages 109–127. Springer International Publishing, Cham.

Petersen-Øverleir A (2004). Accounting for heteroscedasticity in rating curve estimates. Journal of Hydrology 292:173–181.

Petersen-Øverleir, A., and Reitan, T. (2005). Objective segmentation in compound rating curves. Journal of Hydrology, 311(1–4), 188–201.

Reitan, T., and Petersen-Øverleir, A. (2009). Bayesian methods for estimating multi-segment discharge rating curves. Stochastic Environmental Research and Risk Assessment, 23(5), 627– 642.

**Citation**: https://doi.org/10.5194/egusphere-2024-1389-RC1

**Reviewer #2**

**Review of egusphere-2024-1389: Assessment of uncertainties on stage-discharge rating curves: A large scale application to Québec hydrometric network Alain Mailhot, Guillaume Talbot, Samuel Bolduc, and Claudine Fortier**

**Comment # 1** Based on stage discharge data from the Quebec hydrometric network the authors present an analysis of the uncertainty of stage-discharge rating curves. In their manuscript detailed description of i.) the selection of hydrometric sides, ii.) the analysis of the temporal stability/instability of the gauging sides, iii.) the fitting of either one or two rating curves to the gaugings as well as iv.) the fitting of different models to quantify the uncertainty of the established rating curves is given. In general, the paper is scientifically sound and well designed and written. However, there are quite some careless errors (e.g. different naming of the uc-models in fig 6 and in the text, etc – see the technical points raised by my fellow reviewer colleague) which complicates the reading of the paper quite a bit.

**Response**: A careful rereading of the paper was conducted, and all corrections proposed by reviewer #1 were made.

**Comment # 2** One point I would like to be tackled is the fact that generally the highest density in available gaugings is for medium stage levels, whereas it seems that there are less dense recordings at more extreme (low and high) stages – at least that is what I take from the two examples depicted in fig 7. Would the result of the study (e.g. the selected uc-models) change if - at least for the gauges with reasonable number of gaugings – the records would be somehow randomly (and repeatedly) sub-selected, so that the data points are somehow uniformly distributed (e.g. one record every 10 cm)

over the h-range the rating curve is valid for. This maybe could add a further aspect towards the robustness of the selected/assigned uc-models.

**Response**: As shown in Figure 1, most gaugings are realized in spring, during the peak flow associated with snow melt, and during summer/autumn. For operational and safety reasons, gaugings are most likely made under medium to low stage conditions as mentioned by the reviewer. Ideally, as pointed out by the reviewer, we would prefer to have uniformly distributed gaugings across the h-range. However sub-selection, or sub-sampling, would eliminate some gaugings and deprive us of valuable information. Such strategy could be applied only to stations with 'reasonable number of gaugings' and this 'reasonable' number would need to be assessed.

A sentence has been added (line 411-412) about that specific issue.

**Citation**: https://doi.org/10.5194/egusphere-2024-1389-RC2

---

## Author Response (AR2)

**Assessment of uncertainties on stage-discharge raging curves: A large scale application to Québec hydrometric network**

**Alain Mailhot, Guillaume Talbot, Samuel Bolduc, Claudine Fortier**

**Point-by-point reply to editor and reviewers' comments**

A point-by point response to editor and reviewer's comments is provided in the following with corresponding references to the revised version of the manuscript. Since the numbering of the lines changed, we indicate in each case the new line numbers where revised text appears. Please note that the changes in the 'marked' version are indicated in green.

**RC Specific comments**

**Comment # 1**

**Lines 100 – 108**: It is not correct that uncertainties in model parameters of rating curve models are accounted for in residual distributions. The residuals account mainly for the uncertainties in the flow, that is, the measurement error in the flow. Discrepancy between the true flow and the proposed mathematical model ends up in the residuals along with the measurement error in the flow. Uncertainty due to measurement error in stage also ends up in the residuals**.**

Statistically, the residuals can only provide information about the value of the variance of the error terms explained about (assuming constant variance for the error term in the logarithmic transformed discharge rating curve model). The residuals alone cannot provide information about the uncertainties of the parameters of the discharge rating curve model. The structure of the discharge rating curve model and the probability model for the observed discharge and stage pairs are needed. Uncertainty in the model parameters has an effect on rating curve predictions, these predictions will have wider prediction intervals when parameter uncertainty is taken into account.

It would be helpful for the readers to know that uncertainty in the model parameters is not addressed. The reasons for not addressing this uncertainty should be given, e.g., this makes the method more robust or this saves computation time. Please do not state that uncertainty in model parameters is accounted for in residual distributions. Also, please state that uncertainty in the model parameters is not addressed in the proposed method and the reasons for that.

**Response**: The reviewer is right, incorrect statements about uncertainties were made in the manuscript. To comply with the reviewer's comments, the text on lines 103-106 was changed.

**Comment #2**

**2) Lines 134 – 141**: I agree that Mean Square Relative Error (MSRE) is a commonly known formula. However, a minimum justification for using MSRE in the context of fitting rating curves is to note that it has been used before in the literature and appropriate reference or references. The power function is a commonly known and frequently used

model for rating curves. It is introduced in eq. (1) and there a reference is cited. Please consider citing a reference for eq. (2), e.g. Gupta et al. (2009).

Reference: Gupta, H.V., Kling, H. Yilmaz, K.K., Martinez, G.F. (2009). Decomposition of the mean squared error and NSE performance criteria: Implications for improvinghydrological modelling, *J. Hydrol.* 377 (1-2): 80-91. DOI: 10.1016/j.jhydrol.2009.08.003.

**Response** We added the reference to Gupta et al. (2009) on line 136

**RC Technical corrections**

**1) Lines 92 – 95:** I was actually referring to the paper Petersen-Øverleir A (2004). The paper Reitan, T., and Petersen-Øverleir, A. (2006) does not discuss the estimation of the variance or standard deviation of the error terms, so, it should not be used in lines 92 – 95.

**Reference:** Petersen-Øverleir A (2004). Accounting for heteroscedasticity in rating curve estimates. Journal of Hydrology Volume 292, Issues 1–4, 15 June 2004, Pages173-181 (https://doi.org/10.1016/j.jhydrol.2003.12.024)

**Response:** The reference has been changed on line 107 and added in the bibliography (lines 521-522).